# Reducing Uncertainty Through Mutual Information in Structural and Systems Biology

**Vincent D. Zaballa** [1]  **Elliot E. Hui** [1]

## Abstract

Systems biology models are useful models of complex biological systems that may require a large amount of experimental data to fit each model's parameters or to approximate a likelihood function. These models range from a few to thousands of parameters depending on the complexity of the biological system modeled, potentially making the task of fitting parameters to the model difficult - especially when new experimental data cannot be gathered. We demonstrate a method that uses structural biology predictions to augment systems biology models to improve systems biology models' predictions without having to gather more experimental data. Additionally, we show how systems biology models' predictions can help evaluate novel structural biology hypotheses, which may also be expensive or infeasible to validate.

## 1. Introduction

Systems biology models start from basic physical and chemical principles, and gradually build more complicated models, from cell circuits to organ systems, that in the limit of complexity can describe human life (Alon, 2019). Despite modeling from fundamental biological knowledge, these models may still require a great deal of data to accurately predict biological responses, reflecting uncertainty in our biological knowledge. Unfortunately, gathering more data to increase the confidence in model predictions may be infeasible due to high cost. Conversely, when beginning a scientific study one may have too many options to choose from to effectively study their system and may want to use educated priors to begin data collection using principled methods such as Bayesian optimal experimental design (BOED) (Lindley, 1956; Rainforth et al., 2024).

[1]Department of Biomedical Engineering, University of California, Irvine, United States. Correspondence to: Vincent Zaballa <vzaballa@uci.edu>.

*Accepted at the 1st Machine Learning for Life and Material Sciences Workshop at ICML 2024.* Copyright 2024 by the author(s).

Structural biology in the form of protein structure prediction has recently made great strides in predicting structures of single-chain proteins based on years of curated data collected from experiments (Jumper et al., 2021; Baek et al., 2021). Prediction of multi-chain protein complexes quickly followed, providing a new depth of understanding to multimeric protein structures (Evans et al., 2021). These new capabilities provide rich details to help form new biological hypotheses but are limited in scope to static descriptions of biological systems.

We demonstrate a method that utilizes the extrapolation capabilities of systems biology models with the structural accuracy of protein structure predictions. This method can improve systems biology models' predictions and refine hypotheses generated by protein structure prediction tools, with implications in drug development. We provide an example using a model of the Bone Morphogenetic Protein (BMP) pathway with previously-collected experimental data and the structure prediction of a surface receptor complex in the BMP pathway. We also show how this method can help to evaluate new structural biology hypotheses, which can be expensive to evaluate using X-ray crystallography or cryo-electron microscopy (cryo-EM) techniques.

## 2. Background

**Bone Morphogenetic Protein pathway**  The BMP pathway is utilized in cell-cell communication, where homologous BMP ligands produced by one set of cells can act in combinations to elicit a response in another set of cells that express BMP receptors (Antebi et al., 2017). This context-dependent messaging means that the same message sent in the form of a combination of BMP ligands by one cell in the pathway can be interpreted in different ways by cells with different sets and concentrations of BMP receptors. This "promiscuous signaling" can be modeled by mass action kinetics, which effectively describes the competitive binding of BMP ligands to BMP receptor complexes. There are multiple models to describe the BMP pathway and we focus on the "onestep" model

$$A_i + B_k + L_j \overset{K}{\rightleftharpoons} T_{ijk}, \tag{1}$$

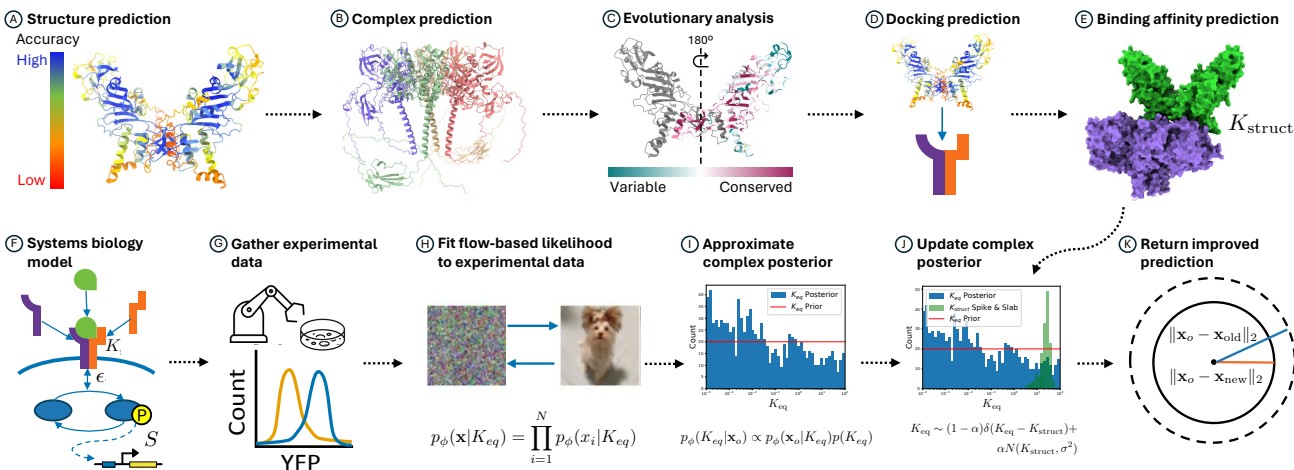

*Figure 1.* Flowchart of our method combining structural (*Top*) and systems biology (*Bottom*) predictions. Top row shows A) initial structure prediction or query from a database, B) prediction of the complex if the structure is not available, C) evolutionary analysis of the sequence, where symmetric proteins such as BMP4 hommodimer ligands only require an analysis from one chain, D) docking prediction, and E) binding affinity prediction of the docked complex ($K_{\text{struct}}$). Bottom row shows F) the initial math model that describes the biological pathway of interest, G) collection of experimental data, H) fitting a flow-based likelihood to the data, I) returning a posterior distribution over model parameters ($K_{\text{eq}}$) for the BMP4-BMPR1A-ACVR2A complex, J) adjusting sampling of the posterior by inclusion of a spike-and-slab distribution of the structurally-predicted binding affinity ($K_{\text{struct}}$), and K) the resulting improved prediction of the systems biology model as measured by median distance from simulated data points using the new posterior. Processes can be run independently up until K).

where the equilibrium constant $K$ represents the steady-state forward and backward reactions, subscripts $i, j, k$ represent the $i^{th}$ type $A$ receptor, the $j^{th}$ BMP homodimer ligand complex, and $k^{th}$ type $B$ receptor, and they all combine in *one step* to form a trimeric complex $T$. The complex then phosphorylates SMAD1/5/8 to then send a downstream gene expression signal (Alarcón et al., 2009). This model can effectively describe how approximately ten homodimeric ligand variants bind with heterotetramer receptor complexes made up of two type I and two type II receptors, of which there are four type I and three type type II receptors. Different combinations of these components help to explain the different tissues that arise during embryonic development (Salazar et al., 2016; Butler & Dodd, 2003) and are implicated in cancer (Bach et al., 2018; Kallioniemi, 2012), making proteins in the pathway a potential drug target. However, even though this model is capable of modeling responses of the BMP pathway (Su et al., 2022), it lacks an explicit likelihood function that can be used to describe the probability of observed data, which is important in uncertainty quantification and BOED. We follow the work of Klumpe et al. (2022) and use this onestep model of the BMP pathway to model 5 BMP ligands, 3 type I, and 2 type II receptors present.

**Simulation-based inference** Simulation-based inference (SBI), also known as Likelihood Free Inference (LFI), relies on simulations from a model $x \sim p(x|\theta)$, observed data $x_o$, and a starting prior over model parameters $p(\theta)$, to fit a probabilistic model of either the likelihood $p(x|\theta)$, posterior

$p(\theta|x)$, or the likelihood-to-evidence ratio $p(x, \theta)/p(x)p(\theta)$ (Cranmer et al., 2020). Normalizing flows are commonly used to approximate a likelihood or posterior given their direct density estimation capabilities (Papamakarios & Murray, 2016; Papamakarios et al., 2018; Greenberg et al., 2019), but diffusion (Sharrock et al., 2022) and flow-matching (Dax et al., 2023) methods can also be used to model the posterior, while classifiers are used to model the likelihood-to-evidence ratio (Gutmann et al., 2018; Brehmer, 2021). SBI methods can be refined over multiple rounds of Bayesian inference, thus refining the likelihood, posterior, or likelihood-to-evidence ratio by repeated use of Bayes' theorem using observed data $p(\theta|x_o) \propto p(x_o|\theta)p(\theta)$. One simple validation metric in SBI is the median distance, which we define as $\text{med}(\|x_0 - x\|_2)$, which is used to measure performance of SBI-based posterior distributions, where the samples drawn from a simulator with an updated posterior should be closer to observed data than samples drawn using the prior. An intuitive way to view this metric is as a hypersphere over the dimension of the data decreasing with new information or better inference methods, as determined by the posterior distribution.

**Protein structure prediction** Protein structure prediction methods such as AlphaFold2 (Jumper et al., 2021), AlphaFold3 (Abramson et al., 2024), and RoseTTAFold (Baek et al., 2021) are capable of producing accurate protein structure predictions given a sequence of amino acids. In addition to predicting single and multi-chain protein structures to high accuracy on their own, these models have been used in

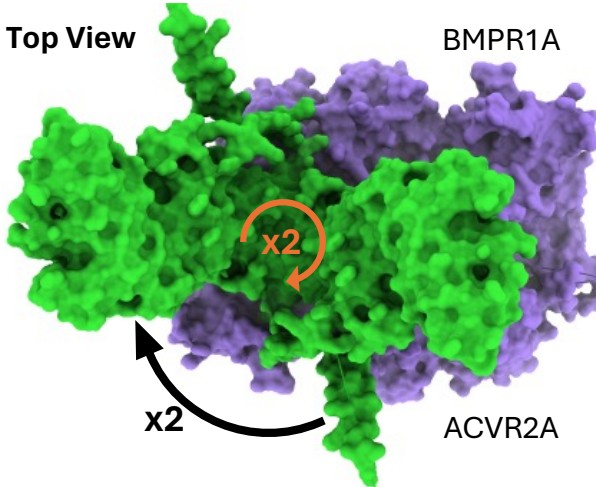

**Top View**      BMPR1A

x2

x2

ACVR2A

*Figure 2.* Symmetries present in the predicted BMP complex. Rotating the BMP4 homodimer (green) 180º about the center of the receptor complex (purple) results in a symmetric binding site as the BMPR1A and ACVR2A receptors have an identical copy mirrored roughly at -45º and 45º about the horizontal and vertical axes, respectively, for two additional binding positions. Rotating the BMP4 homodimer 180º about its own center results in two more positions, for a total of four possible binding positions.

integrative modeling with experimental modalities, such as cryo-EM, to achieve unprecedented accuracy in large-scale protein structure prediction that provides greater insight as to how biological complexes like the nuclear pore complex operate (Fontana et al., 2022).

**Evolutionary analysis** To predict how two protein structures might dock, evolutionary conservation data offers crucial insights. Proteins that interact typically exhibit conserved amino acids at their binding sites, as these regions are crucial for catalysis and interaction (del Sol Mesa et al., 2003). Evolutionary analysis uses a set of homologous sequences, their alignment, and statistical methods to determine a degree of conservation of each amino acid in the sequence of interest. We used the ConfSurf server (Celniker et al., 2013) to identify which amino acids in the BMP homodimer and receptor complex were likely responsible for binding in the complex.

**Protein-protein docking** Prediction of how multiple multichain proteins dock with one another is an important step in understanding how complexes behave, and is a necessary preconditioning task for predicting protein-protein binding affinity. There are many tools for docking (Lyskov & Gray, 2008; Desta et al., 2020) and we used HADDOCK (Van Zundert et al., 2016), a software tool that is able to apply constraints as to which amino acids in the sequences provided are important to docking. Specifically, we apply the evolutionary conservation data to the Ambiguous Interaction Restraints (AIRs) to identify active and passive residues in the protein complex that facilitate binding.

**Protein-protein binding affinity prediction** Interactions between proteins play a critical role in almost all forms of cellular function, from DNA replication to signal transduction. The binding affinity is the measure of how likely a complex will form between two or more proteins, and is thus an important variable in biological systems. The affinity is typically described through the dissociation constant $K_d$, which is related to the Gibbs free energy $\Delta G = RT \ln K_d$. This is a different quantity from the equilibrium constant, $K$, of mass action kinetics and shown in Eq. (1). We can relate the Gibbs free energy to the equilibrium constant of mass action kinetics by the principle of detailed balance by $-\ln \boldsymbol{K} = \mathbf{N}\boldsymbol{G}$, where $\mathbf{N}$ is the stoichiometric matrix of the elements involved in binding (Dirks et al., 2007). We are now able to relate posterior parameter predictions of the mass action chemical equilibrium parameter $K$ with protein binding affinity dissociation constant $K_d$. We distinguish the binding affinity from structural modeling by calling it $K_{\text{struct}}$ for structure-based prediction of binding affinity and $K_{\text{eq}}$ for mass action kinetics-based prediction of binding affinity. We evaluate both results in terms of mass action kinetics's binding affinity, which is negative natural logarithm of the dissociation constant $K_d$, meaning higher $K$ represents stronger binding and lower represents weaker binding. We use Prodigy (Vangone & Bonvin, 2015), a statistical model of binding affinity based on a pre-docked protein complex, to predict binding affinity from a docked structure due to its accuracy and ease of use.

## 3. Related Work

Previous work to fit parameters of the BMP pathway relied on maximum likelihood estimation point estimates (Klumpe et al., 2022). However this point estimate directly lacks a probabilistic interpretation. Bootstrap can provide a pseudo-probability distribution that can be sampled but cannot evaluate the probability of a new data point. Alternatively, there are many methods for predicting protein-protein interactions using graph neural networks, ranging from contact location prediction (Yuan et al., 2022; Gainza et al., 2020) to classification of the interactions between proteins in a graph (Xu et al., 2024). These predictions lack a physics-based relation to the dissociation constant ($K_d$), unlike mass action kinetics, making them incomparable to our method. Hence, we only evaluate how our method increases information in systems biology using structural biology.

## 4. Method

We combine the structural and systems biology components, whose general workflow is shown in Fig. 1, to improve the median distance calculation during SBI. We chose to model the BMP4 homodimer ligand and a receptor complex composed of two BMPR1A and two ACVR2A receptors.

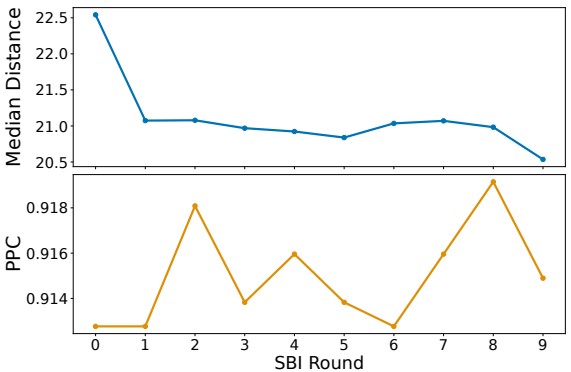 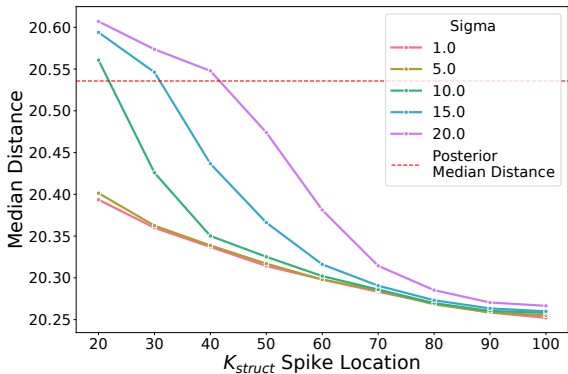

*Figure 3.* (*Left*) Median distance and posterior predictive check (PPC) plots of a trained surrogate normalizing flow likelihood of the BMP model over 9 rounds. Decreasing median distance over multiple rounds indicates improved prediction accuracy and increasing PPC indicates improved flexibility in modeling observed data $\mathbf{x}_o$. (*Right*) Median distance curves over different spike locations and varied by the amount of noise in the slab portion of the spike-and-slab distribution showing improvement in the predicted median distance as the strength of the binding affinity increases. The red horizontal line is the last posterior's median distance from the initial SBI training (*Left*).

**Predicting structures in the BMP pathway & binding affinity prediction** First, we predicted the structure of BMP4 homodimer and the ACVR2A and BMPR1A receptor complex using AlphaFold2 via ColabFold (Mirdita et al., 2022). We used AlphaFold2 multimer for prediction of the BMP4 homodimer and the receptor complex. Once the multi-chain structures were predicted, we performed an evolutionary analysis using the ConfSurf server to identify conserved evolutionary regions that would assist in specifying AIRs during docking. Once we selected the most likely docked structure based on the Haddock scoring function and feasibility of the docked structure, we used the docked complex to predict binding affinity, $K_{\text{struct}}$, using the Prodigy server.

**Posterior prediction of BMP model parameters** We used publicly-available data collected on the BMP pathway to train a normalizing flow surrogate of a likelihood function, $p_\phi(\mathbf{x}|\boldsymbol{\theta})$, where $\boldsymbol{\theta}$ represents all sixty parameters in the BMP onestep model, but we will focus on one parameter $K_{\text{eq}}$ that represents the equilibrium constant of the BMP4-BMPR1A-ACVR2A complex. The data have 940 dimensions $\mathbf{x} \in \mathbb{R}^{940}$, with each dimension having five conditioning variables in addition to the shared $\boldsymbol{\theta} \in \mathbb{R}^{60}$. This can be computationally difficult for normalizing flows, but we circumvent this issue by using the identity of the joint distribution for i.i.d. data: that the joint likelihood can be described by the product of likelihoods for each individual experiment $p_\phi(\mathbf{x}|\boldsymbol{\theta}) = \prod_i^N p_\phi(x_i|\boldsymbol{\theta})$, which we modeled using a neural spline flow (Durkan et al., 2019). We discuss architecture choice and limitations in Appendix B. Our starting prior $p(\boldsymbol{\theta})$ is a log-uniform distribution in the domain $[10^{-4}, 10^2]$. We train the flow by maximum likelihood over nine rounds of sequential updates of SBI on the observed data, $\mathbf{x}_o$, and evaluate its performance on the median dis-

tance metric and its posterior predictive coverage (PPC). PPC is a percent of the data that is covered by simulations from the new posterior, rather than a distance, and we want to cover as much data as possible. We return a posterior by using a variational inference-based flow network that approximates the posterior $q_\psi(\boldsymbol{\theta})$ given its speed over MCMC-based methods and similar accuracy (Glöckler et al., 2022). We provide details about the SBI procedure in Appendix C.

**Integrating structural & systems biology** There are two straightforward ways to incorporate structural information into systems biology. First, as a prior of one of the parameters $\boldsymbol{\theta}$, of the systems biology model, and second as a distribution that is combined with the posterior distribution to return samples from weighted samples from both distributions. We chose the second method as we first fit a posterior, evaluated it for good structural candidates, such as whether the posterior has a clear bimodal distribution, and then updated the posterior distribution with a spike-and-slab distribution (Lempers, 1971). The spike-and-slab distribution models the point information of $K_{\text{struct}}$ as a dirac delta function and adds Gaussian noise in a predetermined amount around it, as well as a mixing parameter where we used $\alpha = 0.1$. Sampling from this model becomes $K_{\text{eq}} \sim (1-\alpha)\delta(K_{\text{eq}} - K_{\text{struct}}) + \alpha N(K_{\text{struct}}, \sigma^2)$. We evaluated samples generated with Gaussian noise of $[1., 5., 10., 15., 20.]$. Finally, we combine the two by drawing a set of initial samples for $K_{\text{eq}}$ from its posterior distribution, $K_{\text{eq}} \sim p(K_{\text{eq}} \mid x)$, then each sampled value of $K_{\text{eq}}$ is then evaluated with the spike-and-slab distribution. Weights are calculated for each sample based on their evaluated probabilities under the spike-and-slab model. These weights are computed to emphasize samples that align well with the spike-and-slab distribution, thereby adjusting the influence of each sample in subsequent analysis. Finally, a

new set of samples for $K_{eq}$ is drawn based on the normalized weights. This resampling emphasizes values that are consistent with both the original data and the assumptions of the spike-and-slab model.

## 5. Results

**Predicting the BMP4-BMPR1A-ACVR2A structure complex** We found for the BMP4 homodimer that the pLDDT (predicted local distance difference test) and PAE (predicted alignment error) were confident for most of the structure except for some exterior components, as shown in Fig. 1 (A). The BMPR1A-ACVR2A receptor complex demonstrated high pLDDT and PAE for components that seem highly probable to function at the surface of the cell, with four alpha helices clearly demonstrating a transmembrane portion of the receptor complex. We show more visualizations of the proteins, show PAE plots, and discuss each multimeric structure prediction in more detail Appendix A. We performed six AlphaFold2 recycles for the BMP4 homodimer and twenty recycles for the receptor complex, and determined the given structures provided sufficient accuracy to proceed to evolutionary analysis. Evolutionary analysis of each complex corroborated the pLDDT and PAE scores, showing that high-confidence areas tended to be conserved while some low-confidence areas that had high conservation subsequently were important for docking the two complexes together. Using the highly-conserved regions of each protein present at the docking interfaces between the two proteins, we were able to dock the two in a non-traditional format. That is, the ligand-receptor complex predicted does not represent the traditional way of how BMP ligands fit with receptor complexes as a single "lock and key" format (Katagiri & Watabe, 2016). Instead, the BMP4 homodimer has a symmetric axis that seems to allow it to dock in two different ways to the same receptor complex, while the receptor complex can be rotated twice to provide a total of four possible binding configurations, as shown in Fig. 2. While our docking procedure only returned one of the four conformations, we took this result into consideration in subsequent analysis. Binding affinity prediction using Prodigy returned a dissociation constant ($K_d$) of $5.8 \times 10^{-12}$ that translates into a binding affinity of $K_{eq} = 10.24$ and when multiplied by four is $K_{eq} = 40.97$, which accounts for the four different ways the complex can be made. We tested this hypothesis in the creation of our spike-and-slab distributions representing the binding affinity, varying the dirac delta values from 20 to 100, the limit of our prior distribution.

**Updating the BMP model posterior with structural information** Fig. 3 shows the result of sampling from the updated distribution on median distance prediction. We can see that the median distance improves as we increase the

prior assumption of the strength of the binding affinity and decrease noise, all the way up to the upper limit of $10^2$ of our prior distribution, $p(\boldsymbol{\theta})$. This is surprising as this indicates that this complex is very tightly binding - more than what our prior modeling assumptions allowed - and warranting a review of our prior modeling assumptions. There is also a discrepancy between the binding affinity produced by Prodigy and the best-performing spike-and-slab distribution. This could be due to error in the Prodigy binding affinity prediction, errors in the structure prediction process, or an error in the systems biology model. However, there is agreement between the systems biology predictions and structural biology predictions indicating that the complex has a strong binding affinity. Regarding the choice of BMP systems biology model, the current onestep model only predicts the binding of BMP ligands binding to pre-formed receptor complexes but BMP ligands are also known to bind to receptor components to induce complex formation (Miyazono et al., 2010). This means the onestep model is conservative in its binding affinity prediction and is a lower bound of possible binding affinities, which is also supported by our data. We did not evaluate the lower ranges of binding affinity as the data suggested a clear worsening trend in the median distance metric the lower the spike was located.

## 6. Discussion

We demonstrated how to leverage structural biology to improve a systems biology model's predictions and in the process supported a new hypothesis of the structure of the BMP4-BMPR1A-ACVR2A complex operation by agreement with a systems biology model's predictions. The benefits of our method are twofold. First, improving systems biology models' predictions can be helpful in cases when it is infeasible to gather more data. By improving predictions of systems biology models, we can improve design of cell circuits in synthetic biology models or development of therapeutics to treat diseases related to those biological pathways. Our demonstration on a single complex in the BMP pathway showed modest improvement in prediction but a more comprehensive inclusion of structural biology data will likely help to improve models' predictions. Second, our approach provides a novel method for proposing and checking structural biology hypotheses by cross validation with a systems biology model, for example supporting the structural hypothesis of a strong binding affinity in the BMP4-BMPR1A-ACVR2A complex, and the structural basis as to why it might have such a strong affinity.

**On structural & systems biology mutual information** We now formalize the use of mutual information in our framework. The mutual information between two random variables $X$ and $Y$ can be defined as

$$I(X;Y) = H(X) - H(X|Y), \qquad (2)$$

which is the change in entropy about one random variable with knowledge of another. Per Theorem 2.6.5 of Cover (1999) (Information Can't Hurt), adding *relevant* conditional information will reduce the entropy of the underlying data distribution, such that $H(X|Y) \leq H(X)$. In our case, let $K_{eq}$ and $K_{struct}$ be latent random variables and $X$ be the random variable of the observed data point, treating the median distance of simulated points $x \sim X$ as a proxy for the entropy. As we demonstrated, $H(X|K_{eq}, K_{struct}) < H(X|K_{eq})$; thus, we are able to increase information gained $I(X; K_{eq}|K_{struct}) > I(X; K_{eq})$. A natural extension of information gain in biology is to drug discovery where models, such as DiffDock (Corso et al., 2022), can be included as conditional information about the binding of a BMP ligand given a certain small molecule drug $p(x|\boldsymbol{\theta}, K_{struct}, \beta)$, where $p(\beta)$ is the probability of a given drug docking and disrupting normal binding.

**Future work** A key concern about this method is the introduction of errors from any component of the structure prediction pipeline, from single-chain structure prediction to binding affinity prediction. Replacing these steps with probabilistic structural models such as AlphaFlow (Jing et al., 2024), protein-protein docking methods, and binding affinity predictions would provide a way to help reduce uncertainty in structure prediction using systems biology data. We leave this for future work.

## Acknowledgements

We would like to thank Heidi Klumpe, Eric Bourgain, and Pieter Derdeyn for helpful discussions. This research was funded by the National Institute of General Medical Sciences (NIGMS) of the National Institutes of Health (NIH) under award number 1F31GM145188-01.

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

## A. Predicted protein complexes

We provide more details about the AlphaFold2 structure prediction errors for both the BMP4 homodimer and BMPR1A-ACVR2A receptor complex in Fig. 4. The errors for the BMP4 homodimer are mainly clustered around the bottom that happened to be the most conserved evolutionary region, which could be due to its active role in binding. The receptor complex is most uncertain in regions within the cell below the four alpha helices that represent a transmembrane region. The receptor's PAE plot indicates it is most confident in the four subunits in the cell surface receptor as well as the four off-diagonal confidences in the pairing of the single-chain units at the cell surface. Since we were most interested in events at the cell surface where confidence was highest, we deemed this an adequate structure prediction.

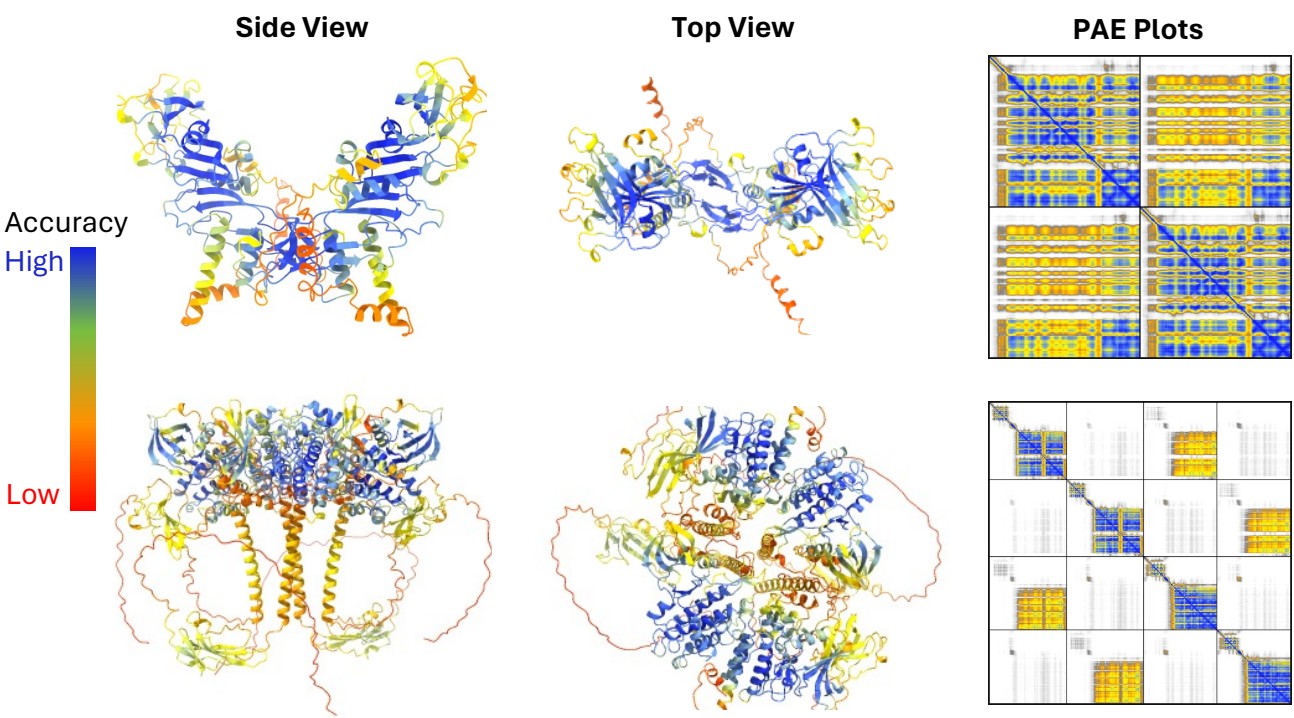

*Figure 4.* AlphaFold2 prediction pLDDT and PAE scores for both the BMP4 homodimer (*Top*) and BMPR1A-ACVR2A receptor complex (*Bottom*).

## B. Normalizing flow architecture design choices & hyperparameters

Normalizing flows model the change of volume from a base to data distribution via $p(x) = p(u)|\det J_T(u)|^{-1}$, where $p(x)$ is the data distribution, $p(u)$ is the base distribution, $T$ is the transformation (also known as a bijector) that must be monotonic and invertible, and $J$ represents the Jacobian of the transformation $T$ at the data point $u$, and where $u = T^{-1}(x)$. We chose to model independent experiments for two reasons. First, in addition to $\boldsymbol{\theta}$, we model conditional experimental information $\xi$ that has 10 dimensions per experiment. Modeling each independent experiment with its corresponding experimental information is more straightforward to optimize in downstream BOED than concatenating or distilling all conditional experimental information as required by modeling the joint distribution. Second, memory requirements for normalizing flows' bijectors scale with $\mathcal{O}(KN^2)$ where $K$ is the number of layers and $N$ is the dimensionality of the data. By modeling the independent distributions we are able to reduce memory burden to $\mathcal{O}(KN)$ since the output is a scalar. Memory efficiency is important in downstream BOED applications where significant computational burden is introduced by contrastive sampling. Hyperparameters for the neural spline flow can be found in Table 1.

*Table 1.* Training hyperparameters.

|  | BMP Model Surrogate Likelihood |
| --- | --- |
| Batch Size | 100 |
| Number of epochs | $10,000$ |
| Learning Rate | $1 \times 10^{-3}$ |
| Hidden Layer Size | 128 |
| Number of Hidden Layers | 2 |
| Number of Flow layers (bijectors) | 5 |
| Number of bins for NSF | 4 |
| Number of SBI Rounds | 9 |

## C. Training SBI

The training procedure for SBI can be seen in Algorithm 1. We train the likelihood by maximum likelihood by maximizing $\mathbb{E}_{\tilde{p}(\boldsymbol{\theta}, \mathbf{x})}(\log p_\phi(\mathbf{x}|\boldsymbol{\theta}))$, where $\tilde{p}(\boldsymbol{\theta}, \mathbf{x}) = p(\mathbf{x}|\boldsymbol{\theta})\tilde{p}(\boldsymbol{\theta})$ represents simulations from an implicit likelihood (the simulator) and draws from the current prior.

---

**Algorithm 1** Sequential Neural Likelihood (SNL) with Posterior Variational Inference

---

1: **Input:** Observed data $\mathbf{x}_o$, estimator $p_\phi(\mathbf{x}|\boldsymbol{\theta})$, number of rounds $R$, simulations per round $N$
2: **Output:** Approximate likelihood $p_\phi(\boldsymbol{\theta}|\mathbf{x}_o)$ and variational posterior $q_\psi(\boldsymbol{\theta})$
3: Set $q_\psi(\boldsymbol{\theta}) = p(\boldsymbol{\theta})$ and $\mathcal{D} = \{\}$
4: **for** $r = 1$ to $R$ **do**
5:     **for** $n = 1$ to $N$ **do**
6:         Sample $\boldsymbol{\theta}_n \sim q_\psi(\boldsymbol{\theta})$
7:         Simulate $\mathbf{x}_n \sim p(\mathbf{x}|\boldsymbol{\theta}_n)$
8:         Add $(\boldsymbol{\theta}_n, \mathbf{x}_n)$ into $\mathcal{D}$
9:     **end for**
10:     (Re-)train $p_\phi(\mathbf{x}|\boldsymbol{\theta})$ on $\mathcal{D}$
11:     (Re-)train $q_\psi(\boldsymbol{\theta})$ by
12:     $\psi^* = \arg\min_\psi D_{KL}(q_\psi(\boldsymbol{\theta}|\mathbf{x}_o) \parallel p(\boldsymbol{\theta}|\mathbf{x}_o))$ with
13:     $p(\boldsymbol{\theta}|\mathbf{x}_o) \propto p(\mathbf{x}_o|\boldsymbol{\theta})p(\boldsymbol{\theta}) \approx p_\phi(\mathbf{x}_o|\boldsymbol{\theta})p(\boldsymbol{\theta})$
14: **end for**
15: **return** $p_\phi(x|\boldsymbol{\theta})$, $q_\psi(\boldsymbol{\theta})$

---