# OpenReview forum: "Reducing Uncertainty through Mutual Information in Structural and Systems Biology"
_ICML.cc/2024/Workshop/ML4LMS — ML4LMS Poster_

### Official Review · Reviewer_Xe3W · 2024-06-12
**Very valuable work to deal with data-shortage in biology**

**Rating:** 9
**Confidence:** 4

**Review:**

## Summary

The paper introduces a method that leverages mutual information to combine structural biology predictions with systems biology models, improving the latter's predictive accuracy without the need for additional experimental data. This approach is demonstrated using the Bone Morphogenetic Protein pathway, showing that the integration of these two biological modeling techniques can refine systems biology models and validate structural biology hypotheses.

## Evaluation

**Quality:** The paper is of high quality, presenting a well-structured and validated method for integrating structural and systems biology models. The methodology is robust, with detailed steps and empirical results supporting the claims.

**Clarity:** The clarity of the paper is good, with comprehensive explanations of the methods and results. However, additional visual aids and detailed examples could further enhance understanding.

**Originality:** The use of mutual information to integrate structural predictions with systems biology models is a novel approach that addresses key challenges in biological modeling.

**Significance:** The work is significant as it offers a practical solution for improving the accuracy of systems biology models without the need for additional experimental data, which is crucial for applications in drug development and synthetic biology.

## Pros and Cons:

**Pros:**

- The proposed method is novel as it combines structural and system biology models using mutual information, enhancing accuracy for prediction.
- The method proposed is a great help to deal with the lack of experimental data in biology.
- The method can be applied to various biological systems and pathways, offering significant utility in biomedical research.

**Cons:**

- Further validation on diverse biological systems and experimental conditions is needed to confirm the method's generalizability.

I would like to suggest the authors to extend the application scenario of this work to genomic data.

---

### Official Review · Reviewer_N6zJ · 2024-06-12
**An elaborated method that combined systems and structural biology to improve median distance prediction over SBI rounds**

**Rating:** 3
**Confidence:** 3

**Review:**

The paper describes a complex multi-stage and elaborated method to combine systems and structural biology systems to improve median distance prediction over SBI rounds. However, it is not clear if a simpler approach might suffice, the experimentation is very limited, no ablation studies have been performed, and no comparison to other methods is presented. As it stands, this paper is not ready for publication. I encourage the authors to further expand the experimental section with ablation studies, additional experiments, and comparison to competing models. With a more thorough evaluation that clearly demonstrates the advantages of the proposed method this paper can pose a meaningful scientific contribution.

See specific questions and comments below:

* Title: where is mutual information being used in this research?

* Method: what are the novel elements of the proposed method?

* Method (Posterior prediction of BMP model parameters): what normalizing flow method was used?

* Figure 3: How meaningful is the median distance improvement  past SBI round 1? (graph looks pretty stagnated, the last drop might not be a trend). Can you show further SBI rounds?

* Figure 3: PPC does not increase in the flow, there is no clear trend.

* Line 188: “The data have 940 dimensions, x ∈ R940, which is computationally difficult for normalizing flows “ - not true, normalizing flow can easily model more dimensions (e.g.,  images)

* Line 191: “using the identity of the joint distribution” - this conditional independence assumption might potentially lead to poor modeling (as opposed to joint modeling of the whole sample). Why is it a reasonable assumption? Did you try to jointly model x?

* Line 195: “we trained the flow over 9 rounds of SBI”
Did you have only 9 data points to use during training? If so, using DL model with normalizing flow might be poor choice given the tiny amount of data.

* Results: the is no comparison to competing methods. How does the proposed method compare to other methods?

* Results: the proposed method includes multi-stages and is rather complex. What is the importance of the modeling choices (e.g., normalizing flow for the posterior, using spike-and-slab distribution)

---

### Official Review · Reviewer_W6zj · 2024-06-12
**Review of Submission 67**

**Rating:** 9
**Confidence:** 3

**Review:**

**Summary**
This paper investigates the effectiveness of incorporating structural biology predictions without involving additional experimental data.

**Strengths**
1. Overall, this paper is well-motivated and will provide valuable insights into the systems biology field.
2. When updating complex posterior distributions, incorporating binding affinity predictions seems reasonable and promising.
3. The proposed future work is well outlined.

**Weaknesses**
1. As the author noted, addressing the initial noise derived from structural predictions is critical. Although harmonizing with existing probabilistic models such as AlphaFlow can be promising, the complexity cannot be overlooked.
2. Regarding practical usage, it would be more comprehensive if the authors included an analysis of time and memory complexity.